# DNMTs Are Involved in TGF-β1-Induced Epithelial–Mesenchymal Transitions in Airway Epithelial Cells

**DOI:** 10.3390/ijms23063003

**Published:** 2022-03-10

**Authors:** Joo-Hoo Park, Jae-Min Shin, Hyun-Woo Yang, Il-Ho Park

**Affiliations:** 1Upper Airway Chronic Inflammatory Diseases Laboratory, Korea University College of Medicine, Seongbuk-gu, Seoul 02841, Korea; pjh52763@korea.ac.kr (J.-H.P.); shinjm0601@hanmail.net (J.-M.S.); yhw444@korea.ac.kr (H.-W.Y.); 2Medical Device Usability Test Center, Korea University Guro Hospital, Guro-gu, Seoul 08223, Korea; 3Department of Otorhinolaryngology-Head and Neck Surgery, Korea University College of Medicine, Seongbuk-gu, Seoul 02841, Korea

**Keywords:** DNA methyltransferase, epithelial–mesenchymal transition, primary nasal epithelial cells, transforming growth factor β-1, tissue remodeling

## Abstract

Chronic rhinosinusitis (CRS) pathogenesis is closely related to tissue remodeling, including epithelial–mesenchymal transition (EMT). Epigenetic mechanisms play key roles in EMT. DNA methylation, mediated by DNA methyltransferases (DNMTs), is an epigenetic marker that is critical to EMT. The goal of this study was to determine whether DNMTs were involved in TGF-β1-induced EMT and elucidate the underlying mechanisms in nasal epithelial cells and air–liquid interface cultures. Global DNA methylation and DNMT activity were quantified. DNMT expression was measured using real-time PCR (qRT–PCR) in human CRS tissues. mRNA and protein levels of DNMTs, E-cadherin, vimentin, α-SMA, and fibronectin were determined using RT–PCR and Western blotting, respectively. DNMT1, DNMT3A, and DNMT3B gene expression were knocked down using siRNA transfection. MAPK phosphorylation and EMT-related transcription factor levels were determined using Western blotting. Signaling pathways were analyzed using specific inhibitors of MAPK. We demonstrated these data in primary nasal epithelial cells and air–liquid interface cultures. Global DNA methylation, DNMT activity, and DNMT expression increased in CRS tissues. DNMT expression was positively correlated with Lund–McKay CT scores. TGF-β1 dose-dependently induced DNMT expression. Further, 5-Aza inhibited TGF-β1-induced DNMT, Snail, and Slug expression related to EMT, as well as p38 and JNK phosphorylation in A549 cells and TGF-β1-induced DNMT expression and EMT in primary nasal epithelial cells and air–liquid interface cultures. TGF-β1-induced DNMT expression leads to DNA methylation and EMT via p38, JNK, Snail, and Slug signaling pathways. Inhibition of DNMT suppressed the EMT process and therefore is potentially a CRS therapeutic strategy.

## 1. Introduction

Chronic rhinosinusitis (CRS), one of the most common chronic inflammatory diseases, is characterized by symptoms such as rhinorrhea, postnasal drip, nasal obstruction or congestion, sinus pain or pressure, and anosmia or hyposmia, lasting for at least 12 weeks [1]. CRS is a multifactorial disease caused by environmental exposure factors, persistent viral infection, and genetic predisposition in its pathogenesis, which are suggested to cause different therapeutic responses [2]. The heterogeneous pathology of chronic sinonasal inflammation in CRS can be differentiated based on the distribution of inflammatory cells, histopathology, and remodeling processes [3]. However, there are limitations in the treatment of CRS based on phenotype. At least 10% of patients with CRS develop recurrent and refractory diseases that do not respond to medical and surgical treatment regimens. Therefore, a new therapeutic approach to treat CRS is required.

Recent research has explored the heterogeneity of CRS using an endotype. Each endotype has various gene expression signatures despite sharing the same genome sequence, largely because of the tightly controlled epigenetic modifications of the noncoding genome [4]. As the inconsistencies in CRS between endotype and phenotype are probably related to epigenetics, epigenetic aberrations may result in complex inflammatory diseases [5]. Epigenetics regulate immune responses and tissue repair and remodeling in various diseases [6]. Epigenetic mutations are increased in CRS patients, compared with the healthy group [7]. Previous studies have reported that DNA methylation contributes to CRS pathophysiology, including an inflammatory response and tissue remodeling. DNA methylation that occurs fifth carbon of the cytosine of the CpG dinucleotide context to form 5-methylcytosine (5-mC) plays an important role in regulating gene transcription and is a representative epigenetic modification. CpG methylation is catalyzed by DNA methyltransferases (DNMTs). DNMT1 is required for the maintenance of all methylation in the genome, while DNMT3A and DNMT3B generally catalyze de novo methylation of either unmethylated DNA or hemi-methylated DNA for maintenance [8]. Recent studies suggest that epigenetic regulation by DNMTs is associated with epithelial–mesenchymal transition (EMT) [9].

Tissue remodeling refers to the structural change that is caused by a persistent inflammatory response. Airway remodeling is characterized by inflammatory cell infiltration, basement membrane thickening, angiogenesis, goblet cell hyperplasia, edema, and fibrosis [10]. Tissue remodeling can be associated with pathological changes in CRS in the respiratory epithelium, lamina propria, and submucosa. Airway remodeling occurs in which epithelial cells lose epithelial properties and assume mesenchymal properties due to persistent inflammatory response in the airway epithelium, which is known as EMT [11]. EMT plays a crucial role in tissue regeneration, organ fibrosis, and wound healing as it is involved in the tissue repair process, which normally converts into fibroblasts and other related cells to reconstruct tissue after trauma and inflammatory injury. However, dysregulated EMT, caused by chronic inflammation, can lead to the destruction of epithelial cell polarity and cell–cell adhesion junction and production of abnormal mesenchymal cells, resulting in pathological remodeling [12]. Recently, it has been suggested that tissue remodeling including EMT is related to disease resistance in CRS. However, the roles and underlying mechanisms of tissue remodeling including EMT in the epithelium of CRS patients are unclear.

We hypothesized that DNA methylation, one of the epigenetic modifications, is related to tissue remodeling, contributing to refractory CRS. DNMTs could be concerned with transforming growth factor (TGF)-β1-induced EMT, which could be involved in tissue remodeling in the upper airway. The goal of the present study was to examine the expression of EMT markers and DNMTs in sinonasal tissues from CRS and control patients. Additionally, EMT features and DNMT expression were evaluated in A549 cells and primary nasal epithelial cells following TGF-β1 stimulation to investigate the role of DNMTs in TGF-β-induced EMT.

## 2. Results

### 2.1. CRS Was Associated with DNA Methylation and DNMT Expression

To identify the relationship between methylation and CRS endotype, we analyzed global DNA methylation and DNMT activity. Global DNA methylation and DNMT activity were significantly increased in CRS without nasal polyp (CRSsNP) and CRS with nasal polyp (CRSwNP), compared with those in the control (Figure 1a,b). The expression levels of DNMT1, DNMT3A, and DNMT3B were significantly increased in CRSsNP and CRSwNP, compared with those in the control (Figure 1c–e). We investigated whether DNMTs expression was associated with severity using the Rhinosinusitis Lund–McKay CT scores. The Lund–Mackay score is a widely used method for the radiological staging of chronic rhinosinusitis by CT scan of the sinus and periosteum complex. The Lund–MacKay score expresses the opacity of the bilateral sinuses on a scale of 0 to 24. DNMT1, DNMT3A, and DNMT3B mRNA levels were positively correlated with the Lund–Mackay CT score (Figure 1f–h). Additionally, we evaluated the expression of markers related to EMT in each group. DNMT expression was negatively correlated with E-cadherin expression, an epithelial marker, and positively correlated with vimentin expression, a mesenchymal marker (Appendix A).

### 2.2. TGF-β1 Induced DNMT Expression in A549 Cells

To determine which stimuli induced DNMT1, DNMT3A, and DNMT3B expression in A549 cells, the cells were treated with various stimuli, such as Toll-like receptor ligands (poly(I:C) and LPS), Th1 cytokines (IFNγ and TNFα), Th2 cytokine (IL-13), pro-inflammatory cytokine (TGF-β1), and IL-1β. Among these cytokines, the treatment of TGF-β1 caused the greatest increase in DNMT1, DNMT3A, and DNMT3B mRNA and protein levels (Figure 2a,b). We evaluated the effect of TGF-β1 on the expression of DNMT1, DNMT3A, DNM3B mRNA, and protein at various concentrations and demonstrated that TGF-β1 induced the expression of DNMT1, DNMT3A, and DNM3B in a dose-dependent manner. (Figure 2c,d). TGF-β1 also increased DNMT activity and global DNA methylation (Figure 2e,f).

### 2.3. Inhibition of DNMT Expression Suppressed the EMT Process in A549 Cells

We determined the effect of DNMTs on the EMT process. TGF-β1 induced EMT in nasal epithelial cells. The cells were treated with 5-Aza (10 μM), a DNMT inhibitor, before pretreatment of A549 cells with TGF-β1. Increased expression of DNMT mRNA and protein induced by TGF-β1 was inhibited by 5-Aza (Figure 3a,b). In addition, 5-Aza also reversed the TGF-β1-induced suppression of E-cadherin expression and induction of α-SMA, vimentin, and fibronectin expression (Figure 3c,d).

### 2.4. Knockdown of DNMTs Inhibited the EMT Process in A549 Cells

To verify the role of DNMTs in EMT induced by TGF-β1 in A549 cells, siRNA specific for each of the three DNMTs (siDNMT1, siDNMT3A, and siDNMT3B) were used to knock down DNMTs. Each siRNA for the DNMTs knocked down the expression of each DNMT (Figure 4a–f). We analyzed whether the DNMT knockdown suppressed the EMT induced by TGF-β1 using real-time PCR and Western blotting. Therefore, the transfection of siDNMT1, DNMT3A, or DNMT3B reversed the TGF-β1-induced suppression of E-cadherin expression and induction of α-SMA, vimentin, and fibronectin expression (Figure 4g–l). These results suggest that DNMT methylation is involved in TGF-β1-induced EMT in A549 cells.

### 2.5. TGF-β1 Induced DNMT Expression through the ERK and p38 Signaling Pathway

TGF-β1 activates both Smad and non-Smad signals, resulting in the EMT process involved in tissue remodeling. It is well known that the MAPK pathway, a representative non-Smad signal, is implicated in regulating the EMT process. To determine whether the MAPK signaling pathway is involved in TGF-β1-induced DNMT expression, resulting in the EMT process, we used three MAPK inhibitors before pretreatment with TGF-β1 in A549 cells. Increased expression of DNMT mRNA and protein by TGF-β1 was inhibited by treatment with the p38 (SB203580) and JNK inhibitor (SP600125) but not the ERK inhibitor (U0126) (Figure 5a,b). DNMT activity and global DNA methylation were increased by TGF-β1 treatment, whereas they were significantly reduced by the p38 and JNK inhibitors (Figure 5c,d).

### 2.6. Inhibition of DNMT Expression Suppressed the EMT Process through the EMT-Related Transcription Factors

To validate whether DNMTs were involved in the EMT process through the EMT-related transcriptional factors, such as Snail and Slug, we treated cells with 5-Aza before treatment with TGF-β1 and then checked the expression of Snail and Slug. TGF-β1 induced Snail and Slug mRNA and protein expression, and 5-Aza pretreatment blocked the effect of TGF-β1 on Snail and Slug expression (Figure 6a,b). Knockdown of DNMTs with siDNMT1, siDNMT3A, or siDNMT3B transfection also inhibited TGF-β1-induced Snail and Slug expression (Appendix A). The effect of Snail and Slug knockdown on TGF-β1-induced DNMT expression in A549 cells was determined. Knockdown of Snail or Slug with the transfection of siSnail or siSlug inhibited DNMT expression (Figure 6c–f). Knockdown of Snail or Slug also reversed the TGF-β1-induced suppression of E-cadherin expression and induction of α-SMA, vimentin, and fibronectin expression (Figure 6g–j).

### 2.7. Inhibition of DNMT Expression Suppressed the TGF-β1-Induced EMT Process in Human Primary Nasal Epithelial Cells and Air–Liquid Interface Cell Cultures

To verify the effect of 5-Aza on the TGF-β1-induced EMT in human primary nasal epithelial cells, we treated TGF-β1-stimulated human primary nasal epithelial cells with 5-Aza. The expression of DNMTs, α-SMA, fibronectin, and collagen type 1 was increased by treatment with TGF-β1. However, this effect was reversed by 5-Aza (Figure 7a–c). The air–liquid interface (ALI) cell culture data showed similar results in primary nasal epithelial cell data (Figure 7d). These data indicated that inhibition of DNMT expression could prevent EMT-associated protein expression in nasal epithelial cells, thereby altering their expression and changing the cellular morphology.

## 3. Discussion

Herein, we demonstrated that global DNA methylation was increased in tissues from CRS patients. We also observed that the expression and activity of DNMT1, DNMT3A, and DNMT3B were increased in patients with CRS. In an in vitro study using A549 cells and primary nasal epithelial cells, TGF-β1 induced the expression of DNMTs. We showed that the inhibition of DNMTs with 5-Aza attenuated the expression of DNMTs and EMT induced by TGF-β1. Silencing of DNMTs also showed similar effects on TGF-β1 induced changes in EMT markers. In signaling molecules related to TGF-β1–DNMTs–EMT pathways, we demonstrated that the ERK, p38, Snail, and Slug signaling pathways were involved in the TGF-β1-induced expression of DNMTs and EMT. These results were reproduced in the experiments using ALI.

Tissue remodeling is a dynamic process involving the reorganization of existing tissues. Regarding physiological status, tissue remodeling is a normal, endogenous process to maintain tissue homeostasis [13]. However, a persistent inflammatory response causes pathologic remodeling, resulting in irreversible structural changes that could explain the recalcitrance of several chronic inflammatory diseases. Moreover, recent data have shown that remodeling is a result of chronic inflammation and also one of the main causes of the progression of chronic inflammatory diseases because it begins early in the development of diseases; therefore, it plays an independent role in the progression of diseases [14,15]. Several chronic upper airway inflammatory diseases, such as asthma, chronic obstructive pulmonary disease, and idiopathic pulmonary fibrosis, are known to be closely related to EMT [16]. CRS, a representative chronic upper airway disease, is also closely related to tissue remodeling. Its importance is widely investigated in recalcitrant cases. In CRS, tissue remodeling appears in various forms, including EMT, subepithelial fibrosis, goblet cell hyperplasia, hypertrophy of bronchial myocytes, submucosal angiogenesis, and increased extracellular matrix deposition, depending on the type of inflammation [10]. Among them, EMT is a consistent phenomenon in the mucosal epithelium of CRS regardless of the type of inflammation or presence of nasal polyps. During EMT, the epithelial marker protein E-cadherin is downregulated, whereas the mesenchymal markers N-cadherin and Snail are upregulated. Although the association between EMT markers and clinical parameters in CRS has not been investigated in depth, preliminary studies have shown that EMT is observed in both phenotypes, and mesenchymal markers (vimentin) are overexpressed and correlated with disease severity (Lund–Mackay computed tomography score) and tissue inflammation. In a previous study, we showed both the involvement of EMT processes in CRS and the correlation between the degree of EMT and the severity of CRS [17].

TGF-β1, secreted by various cell types, including immune cells and structural cells, is a multifunctional cytokine. TGF-β1 regulates proliferation, differentiation, adhesion, migration, and other cellular functions and is involved in the pathogenesis of a variety of airway diseases related to remodeling, including CRS [18]. Several studies have reported that the expression of TGF-β1 is increased in the nasal mucosa of CRSsNP and early-stage polyps, compared with that in normal tissue [19,20]. TGF-β1 induces EMT processes that downregulate tight junction proteins, such as E-cadherin and ZO-1, and upregulate expression of mesenchymal markers, such as α-SMA, fibronectin, and collagen [21]. TGF-β1 elicits its cellular responses by forming a ligand-induced complex formation of TGF-β type I and type II cell surface receptors. In this study, we observed that DNMTs were involved in TGFβ1-induced EMT in nasal epithelial cells and that the effects of DNMTs were related to p38 and JNK in the TGF-β1 signaling pathway. p38 and JNK, members of the MAPK family, are activated by most stimuli such as inflammatory cytokines, while ERK1 and ERK2 are preferentially activated by mitogen stress. Phosphorylation of ERK, another non-Smad MAPK signaling molecule, was not suppressed by the inhibition of DNMTs. We hypothesized that DNMTs exert their effects through JNK and p38. Well-known EMT-related transcription factors, such as Snail and Slug, were also involved in the TGF-β1–DNMT–EMT pathway. We demonstrated that blocking DNMT expression with siRNA and 5-Aza inhibited the expression of EMT-related transcription factors and vice versa. Although our results could not reveal a clear upper–lower relationship, it was established that they were closely related to each other. It is a limitation that we confirmed the above mechanism using A459 and confirmed it using primary nasal epithelial cells. Due to difficulty in both obtaining the relevant target organ tissues and culturing them, most upper respiratory epithelial studies have relied heavily on commercially produced cells or transformed cell lines, including A549.

In this study, we observed that global DNA methylation and DNMT expression were increased in CRS patients and that DNA methylation was involved in TGF-β1-induced EMT in nasal epithelial cells. We showed that DNMT3A seems to be the most affected by TGF-β1 and inhibitors. the modulation of DNMT3A, de novo DNA methyltransferase, is believed to have a high correlation with the TGF-induced EMT process. Various studies have shown that CRS and DNA methylation are closely related to each other in various ways. It is known that changes in gene expression caused by methylation can lead to cellular overgrowth and widespread inflammation in the respiratory tract [22]. In the nasal mucosal tissues of CRS patients, 2848 genes showed differential methylation, which was mainly functionally characterized as being associated with inflammation [23]. In a report related to DNA methylation associated with CRS, differences in DNA methylation were observed between the eosinophilic CRSwNP and non-eosinophilic CRSwNP, resulting in different pathological characteristics [24]. Additionally, the promoters of the COL18A1, PTGES, PLAT, and TSLP genes are hypermethylated in CRSwNP, compared with those in the controls [25]. In CRSsNP, DNA methylation of the proximal promoter of IL-8 was identified in nasal epithelial cells of patients. EMT, the target pathologic phenomenon of this study, is also related to hypermethylation. DNA hypermethylation at the promoter region of Cdh1 resulted in silencing of E-cadherin and induction of EMT [26]. Dynamic and reversible changes in DNA methylation, regulated by DNMTs, were observed during the TGF-β1-induced EMT process, which could be expected to contribute to the pathogenesis and recalcitrance of diseases. Based on the above findings, we speculate that DNMT inhibitors suppress the progression of CRS pathology by regulating DNA methylation.

It has been found that 5-Aza causes DNA demethylation or hemi-demethylation that regulates gene expression by “opening” the chromatin structure, which is detectable as increased nuclease sensitivity. Moreover, 5-Aza stimulates differentiation and apoptosis of leukemia cells and is used to treat acute myeloid leukemia, chronic myelomonocytic leukemia, myelodysplastic syndrome, and refractory anemia [27]. Additionally, research on the potential of 5-Aza as a therapeutic agent in various diseases related to tissue remodeling is actively conducted. In this study, we observed that DNA demethylation using 5-Aza yielded an anti-EMT effect in nasal epithelial cells, demonstrating the potential of 5-Aza as a candidate drug to treat recalcitrant CRS. However, this drug has been reported to induce excessive cytotoxicity at high doses [28]. Therefore, to proceed with further studies, it is necessary to determine a way to reduce the toxicity of the drug. We expect that local treatment using nasal spray will be a good alternative to oral administration.

## 4. Materials and Methods

### 4.1. Patients and Specimens

Normal uncinate processes were collected from patients who underwent transsphenoidal pituitary tumor surgery. Uncinate process and nasal polyp specimens were obtained from patients of CRS who underwent endoscopic sinus surgery at the Department of Otorhinolaryngology—Head and Neck Surgery of Korea University Guro Hospital. The diagnostic criteria for CRS patients are classified according to guidelines of the 2012 European position paper on rhinosinusitis and nasal polyps [29]. Patients who used systemic or topical steroids, macrolides, and antibiotics for at least 4 weeks before surgery and patients with other upper respiratory tract diseases were excluded. The clinical characteristics of the patients are summarized in Appendix A. In order to perform this study, approval was obtained from Korea University Medical Center Institutional Review Board (Approval Number: 2020GR0308), and written informed consent was obtained from all patients in accordance with the Declaration of Helsinki. The clinical characteristics of the patients are described in Appendix A.

### 4.2. Cell Cultures

Human airway epithelial cell line A549 (ATCC CCL-185) was purchased from the American Type Culture Collection (ATCC, Rockville, MD, USA) and cultured in RPMI 1640 medium (Hyclone, Logan, UT, USA) containing 10% fetal bovine serum (FBS, Gibco, Erie County, NY, USA), 1% penicillin–streptomycin (Invitrogen, Carlsbad, CA, USA) in a humidified incubator at 37 °C and 5% CO_2_. The primary nasal epithelial cells were collected by scraping the mid-inferior turbinate with a brush and were immediately placed in RPMI 1640 medium containing 1% penicillin–streptomycin. After centrifugation at 1500 rpm for 3 min, the red blood cells were removed using RBC lysis buffer (Sigma-Aldrich, St. Louis, MO, USA). Human primary nasal epithelial cells were cultured in PneumaCult-Ex Plus Medium (STEMCELL Technologies, Vancouver, BC, Canada) for 3 days in collagen-type-1-coated dishes (Corning Incorporated, Corning, NY, USA).

### 4.3. Global Methylation Assay

Genomic DNA was extracted using a genomic DNA isolation kit (Epigentek, Nassau County, NY, USA) from nasal mucosa tissue and cells according to manufacture instructions. Quantification of global DNA methylation in extracted genomic DNA was measured by MethylFlash™ Global DNA Methylation (5-mC) ELISA Easy Kit (Epigentek). Briefly, each 100 ng of genomic DNA was added to each assay well to bind DNA and incubate at 37 °C for 60 min. After washing, 5-mC Detection Complex Solution was added to each well for reaction with DNA methylation (5-mC). Color developer solution and stop solution were added sequentially, and when the color change stopped, the absorbance was immediately measured using a microplate reader (Bio-Rad, Hercules, CA, USA).

### 4.4. DNMT Activity Assay

The activity of DNMT was measured using EpiQuik™ DNMT Activity/Inhibition. ELISA Easy Kit (Epigentek) according to manufacture instructions. The nuclear extracts from nasal mucosa tissue and cells were isolated using the Nuclear Extraction Kit (Epigentek). Prepared nuclear extracts incubated onto microplate wells coated with a universal DNMT substrate. Detection complex solution was added to each well to help antibody binding and signal enhancing. After adding color developer solution and stop solution, the absorbance was read on a microplate reader at 450 nm with an optional reference wavelength of 655 nm.

### 4.5. Real-Time PCR

The total RNA was isolated from nasal mucosa tissue and cells using the TRIzol reagent (Invitrogen) according to the manufacturer’s instructions. cDNAs were synthesized from 2 μg of RNA using M-MLV reverse transcriptase (Invitrogen) and oligo (dT). The synthesized DNA and primers of target RNA (Appendix A) were mixed with Power SYBR Green PCR Master Mix (Applied Biosystems) to amplify the gene, and then amplification products were measured using Quantstudio3 (Applied Biosystems, Foster City, CA, USA). The target mRNA expression was normalized with target genes/housekeeping gene (GAPDH) mRNA ratios.

### 4.6. Western Blot Analysis

A549 and primary nasal epithelial cells were lysed using RIPA buffer (Cell Signaling Technology, Danvers, MA, USA) for isolation of protein extracts. The extracted protein concentrations were quantitated using the Bradford assay reagent (Bio-Rad). The equal amounts of protein (20 μg) from each sample with 5X SDS–PAGE loading buffer (Biosesang, Seongnam-si, Korea) were separated by 10% SDS–PAGE gel and transferred onto polyvinyl difluoride membranes (Millipore Inc., Billerica, MA, USA). The membranes were placed in 5% skim milk to block nonspecific binding for 1 h at room temperature. After washing with TBST (Tris-buffered saline, 0.1% Tween 20), the membranes were incubated with primary antibodies overnight at 4 °C. The primary antibodies included anti-DNMT1 (1:1000), anti-DNMT3A (1:1000), anti-DNMT3B (1: 1000, NOVUS Biologicals, Oakville, ON, Canada), anti-α-SMA (1:1000), (Abcam, Cambridge, MA, USA), anti-fibronectin (1:1000), anti-GAPDH (1:1000, Santa Cruz Biotechnology, Inc., Santa Cruz, CA, USA), anti-vimentin (1:1000), anti-Snail (1:1000), and anti-Slug (1:1000, Cell Signaling Technology, Danvers, MA, USA). Following incubation of primary antibody, HRP-conjugated anti-mouse or anti-rabbit antibodies (Vector Laboratories, Burlingame, CA, USA) were added to the membrane for 1 h. Immunoblot signals representing the amount of specific protein were detected with the ECL system (Pierce, Rockford, IL, USA).

### 4.7. Short Interfering RNA Transfection

DNMT1, DNMT3A, and DNMT3B short interfering (si)RNA (Bioneer, Daejeon, Korea) were used to silence or knock down the expression of DNMT1, DNMT3A, and DNMT3B. Each siRNA or negative control siRNA (SN-1013, Bioneer) was mixed with LipofectamineTM RNAiMax (Invitrogen) in Opti-MEM (Gibco). The cells were treated with the mixtures of siRNA–Lipofectamine and incubated at 37 °C and 5% CO2. After 24 h, the medium containing mixtures of siRNA–Lipofectamine was removed, and the medium containing 10% FBS was replaced in transfected cells.

### 4.8. Immunofluorescence Staining

To perform immunofluorescence staining, the cells were fixed with 4% paraformaldehyde for 30 min and permeabilized with 0.1% Triton X-100 (Sigma) for 10 min. After washing three times with TBS-T, cells were blocked with 3% bovine serum albumin for 1 h. Cells were incubated with primary antibodies anti-DNMT1, anti-DNMT3A, anti-DNMT3B, anti-E-cadherin (1:200) diluted in 3% BSA overnight at 4 °C. Additionally, then the cells were incubated with secondary antibodies for 1 h. For secondary antibodies, Goat anti-Mouse IgG (H + L) Alexa 488 or Goat anti-Rabbit IgG (H + L) Alexa 555 (Invitrogen; 1:200) were used depending on the host of the primary antibodies. Nuclei were counterstained using 4’-6-diamidino-2-phenylindole (DAPI, Invitrogen). Fluorescence images were acquired using a confocal laser scanning microscope LSM700 (Zeiss, Oberkochen, Germany).

### 4.9. Statistical Analysis

The data are presented as the means  ±  standard deviation of experiments, repeated at least three times, and analyzed by GraphPad Prism 5 (Graph Pad Software 8.0, San Diego, CA, USA), with unpaired *t*-tests or one-way analysis of variance, and Tukey’s test. Each experiment was conducted in triplicate. *p* values < 0.05 were considered statistically significant difference.

## Figures and Tables

**Figure 1 ijms-23-03003-f001:**
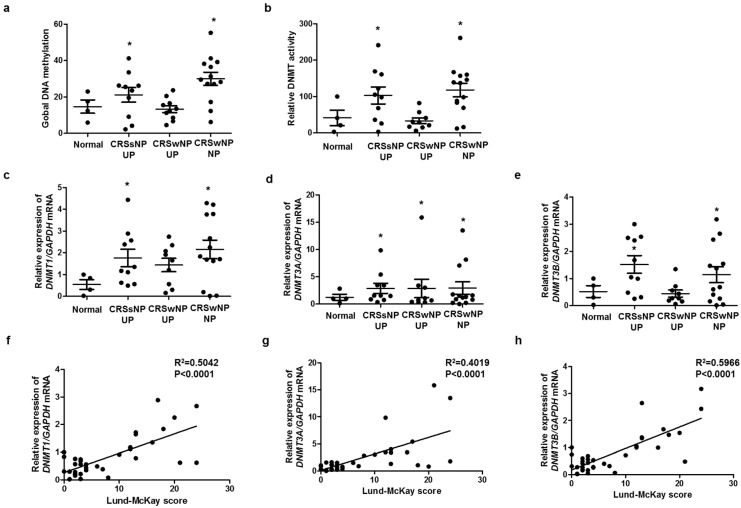
DNA methylation and DNMT expression in nasal tissues: (**a**) global DNA methylation and (**b**) DNMT activity were analyzed using a Global DNA Methylation Assay Kit and DNMT Activation Assay Kit in nasal mucosa tissue from healthy controls and CRS patients. Global DNA methylation and DNMT activity were significantly increased in CRSsNP and CRSwNP, compared with those in the control; (**c**–**e**) DNMT1, DNMT3A, and DNMT3B mRNA expressions were analyzed using real-time PCR in nasal mucosa tissue from healthy controls and CRS patients; (**f**–**h**) the Lund–McKay score was correlated with DNMT1, DNMT3A, and DNMT3B expression in nasal mucosa tissue. UP: uncinate process, NP: nasal polyp. Data are expressed as the mean ± SEM. * *p* < 0.05 vs. control.

**Figure 2 ijms-23-03003-f002:**
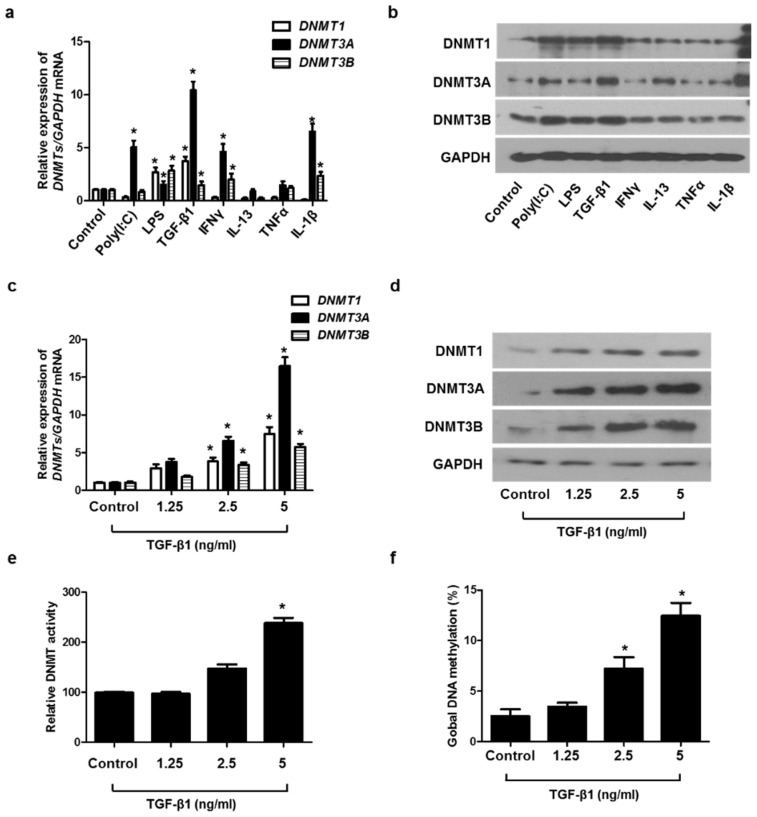
TGF-β1-induced DNMT expression in A549 cells: (**a**) DNMT mRNA and (**b**) protein expression were measured using real-time PCR and Western blotting, respectively, in A549 cells after stimulation with Poly IC (10 μg/mL), LPS (10 ng/mL), TGF-β1 (5 ng/mL), IFN-γ (100 ng/mL), IL-13 (10 ng/mL), TNF-α (50 ng/mL), and IL-1β (10 ng/mL) for 24 h; (**c**) real-time PCR analyses of DNMT1, DNMT3A, and DNMT3B revealed dose-dependent effects in mRNA expression following TGF-β1 treatment; (**d**) protein expression of DNMT1, DNMT3A, and DNMT3B with TGF-β1 treatment in a dose-dependent manner was measured using Western blotting; (**e**) DNMT activity and (**f**) global DNA methylation with TGF-β1 treatment in a dose-dependent manner were detected using a Global DNA Methylation Assay Kit and DNMT Activation Assay Kit. Data are expressed as the mean ± SEM of three independent experiments using a single A549 cell line. * *p* < 0.05 vs. control.

**Figure 3 ijms-23-03003-f003:**
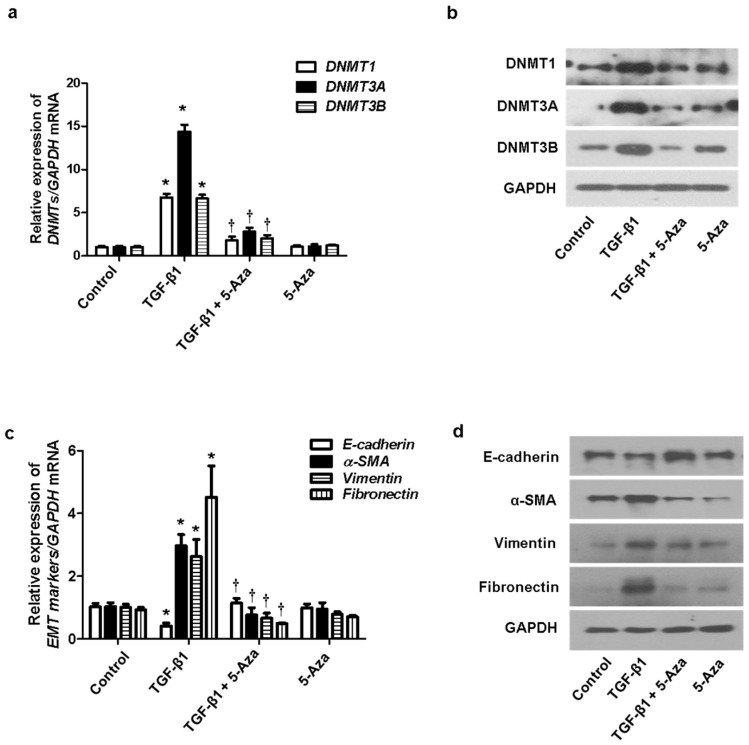
Effect of 5-Aza on EMT induced by TGF-β1 in A549 cells. A549 cells were stimulated with TGF-β1 (5 ng/mL) with or without the DNMT inhibitor (5-Aza, 10 μM): (**a**) the mRNA expression levels of DNMT1, DNMT3A, and DNMT3B were determined using RT–PCR; (**b**) protein expression levels of DNMT1, DNMT3A, and DNMT3B were measured using Western blotting; (**c**,**d**) effects of 5-Aza on the mRNA and protein levels of E-cadherin, vimentin, α-smooth muscle actin protein, and fibronectin expression in TGF-β1-stimulated A549 cells as determined using RT–PCR and Western blotting. Data are expressed as the mean ± SEM of three independent experiments. Values are expressed as the mean ± SEM of three independent samples using a single A549 cell line. * *p* < 0.05 vs. control; † *p* < 0.05 vs. TGF-β1 alone.

**Figure 4 ijms-23-03003-f004:**
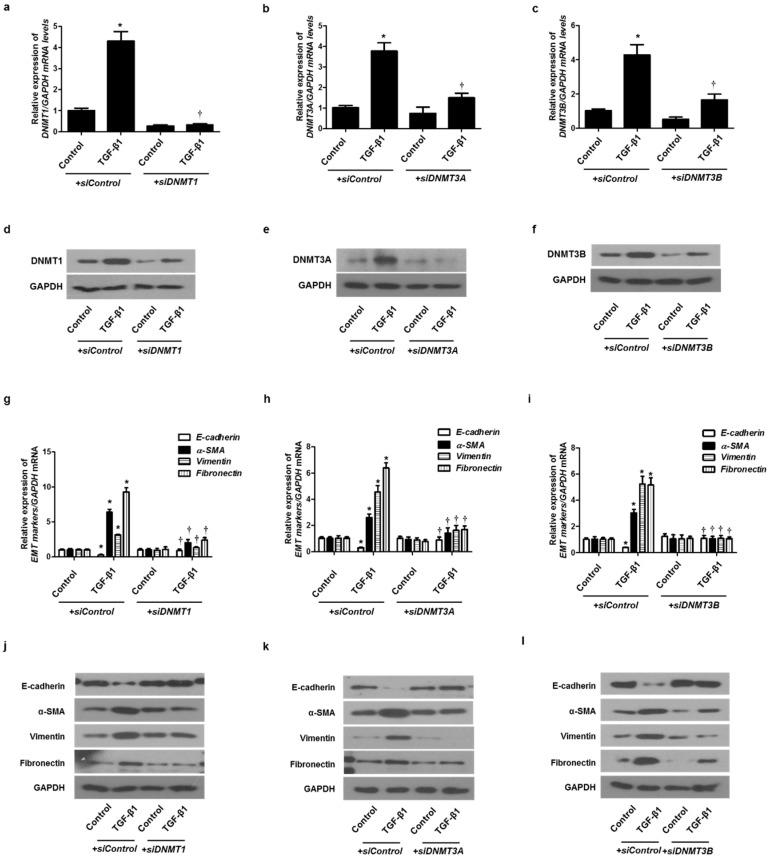
TGF-β1-induced EMT in A549 cells was inhibited by *DNMT1*, *DNMT3A*, and *DNMT3B* siRNAs. Specific *DNMT1*, *DNMT3A*, and *DNMT3B* siRNAs were transfected before treatment with or without TGF-β1 for 24 h for mRNA expression and 72 h for protein expression in A549 cells: (**a**–**c**) mRNA levels of DNMT1, DNMT3A, and DNMT3B were determined using real-time PCR; (**d**–**f**) protein levels of DNMT1, DNMT3A, and DNMT3B were determined using Western blotting; (**g**–**i**) mRNA levels of EMT-related markers (E-cadherin, vimentin, α-smooth muscle actin protein, and fibronectin) were measured using real-time PCR after transfection of DNMT1, DNMT3A, or DNMT3B siRNA with or without TGF-β1 for 24 h; (**j**–**l**) Protein levels of EMT-related markers were measured using Western blotting after transfection of DNMT1, DNMT3A, or DNMT3B siRNA with or without TGF-β1 for 72 h. Data are expressed as the mean ± SEM of three independent experiments using a single A549 cell line. * *p* < 0.05 vs. control with siControl; † *p* < 0.05 vs. TGF-β1 with siControl.

**Figure 5 ijms-23-03003-f005:**
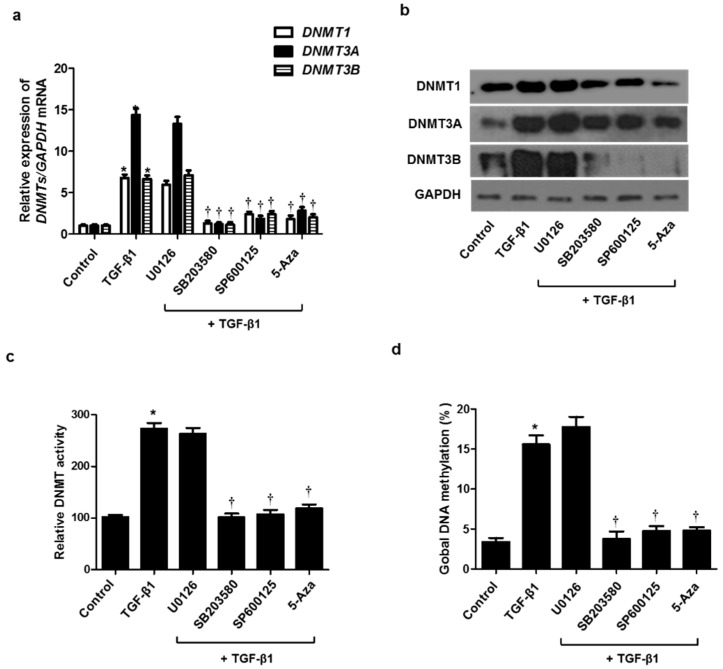
Signaling pathways of TGF-β1 on DNMT expression. A549 cells were pretreated with ERK inhibitor (10 µM U0126), p38 inhibitor (10 µM SB203580), JNK inhibitor (10 µM SP600125), and 5-Aza for 1  h, and then stimulated with TGF-β1: (**a**) mRNA and (**b**) protein expression levels were determined using real-time PCR and Western blotting; (**c**) DNMT activity and (**d**) global DNA methylation were detected using the Global DNA Methylation Assay Kit and DNMT Activation Assay Kit, respectively. Data are expressed as the mean ± SEM of three independent experiments using a single A549 cell line. * *p* < 0.05 vs. control; † *p* < 0.05 vs. TGF-β1 alone.

**Figure 6 ijms-23-03003-f006:**
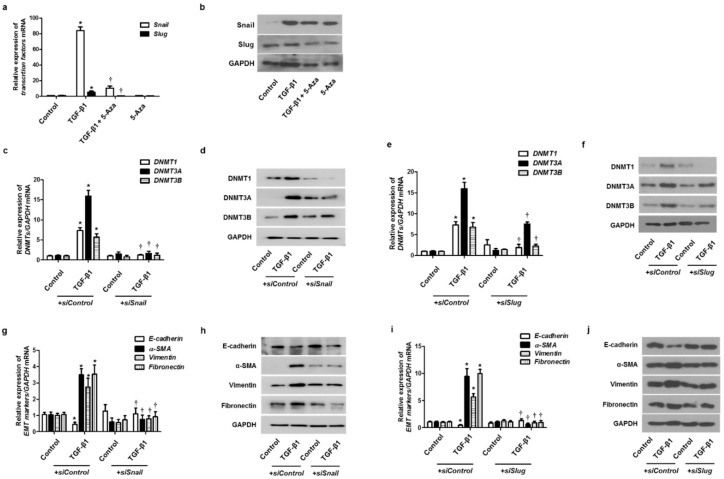
EMT-related transcription factor with TGF-β1 treatment on DNMT expression. A549 cells were pretreated with 5-Aza before TGF-β1 treatment: (**a**) mRNA and (**b**) protein levels of Snail and Slug were determined using real-time PCR and Western blotting, respectively, normalized to GAPDH. Specific Snail and Slug siRNAs (100 nM) were transiently transfected before treatment with or without TGF-β1 (5 ng/mL) for 24 and 72 h; (**c**–**j**) expression of DNMT and EMT-related markers was determined using real-time PCR and Western blotting. Data are presented as mean  ±  SEM. Results were obtained from at least three independent experiments using a single A549 cell line. * *p*  <  0.05 vs. control; † *p*  <  0.05 vs. TGF-β1 with siControl.

**Figure 7 ijms-23-03003-f007:**
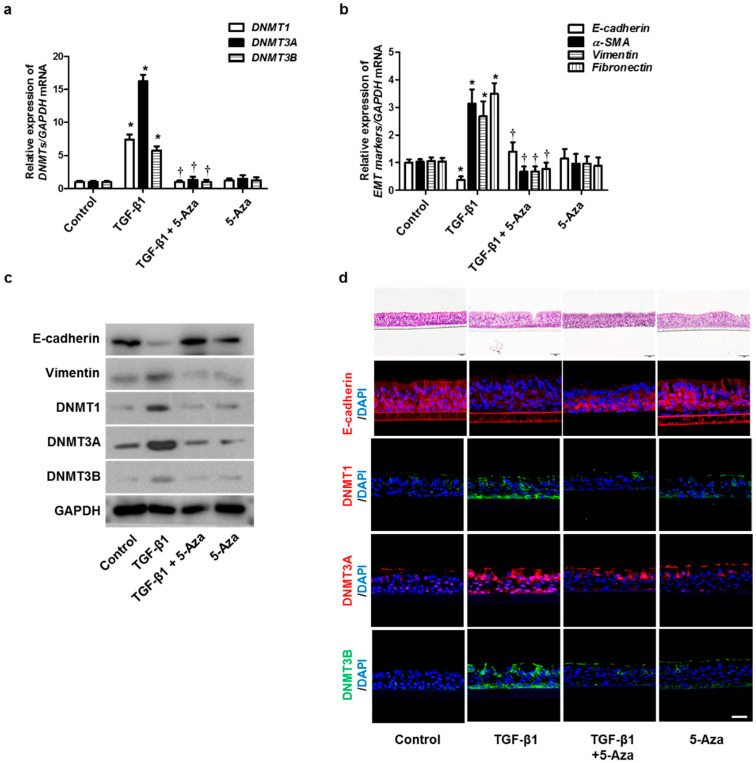
Effects of 5-Aza on EMT induced by TGF-β1 in primary nasal epithelial cells and air–liquid interface culture. Human primary nasal epithelial cells were stimulated with TGF-β1 (5 ng/mL) with or without the DNMT inhibitor (5-Aza, 10 μM): (**a**,**b**) mRNA expression levels of DNMT1, DNMT3A, DNMT3B, and EMT-related markers were determined using RT–PCR; (**c**) protein expression levels of DNMT1, DNMT3A, DNMT3B, and EMT-related markers were measured using Western blotting; (**d**) effects of 5-Aza on E-cadherin, DNMT1A, DNMT3A, and DNMT3B expression in TGF-β1-stimulated human primary nasal epithelial cells were determined using immunofluorescence. Values are expressed as the mean ± SEM of three independent samples using primary nasal epithelial cells isolated from three different CRS patients. * *p* < 0.05 vs. control; † *p* < 0.05 vs. TGF-β1 alone.

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
