# Peer review of "DNMTs Are Involved in TGF-β1-Induced Epithelial–Mesenchymal Transitions in Airway Epithelial Cells"

_ijms, 2022, doi:10.3390/ijms23063003_

Round 1

Reviewer 1 Report

The article titled “The DNMTs are involved in TGF-β1-induced epithelial–mesenchymal transitions in airway epithelial cells” has addressed the possible mechanism by which CRS pathogenesis occurs mediated by epigenetic molecules, DNMTs via TGF-β1. The authors have been able to show how DNMT molecules are induced by TGF-β1 in the tissue samples analyzed and also in cells and its impact on EMT-markers. 

Major concerns

  • In some experiments 1 ng/ml TGF-β1 was used and in others 5 ng/ml with no proper justification.
  • Statistical significance was obtained but the test used for each result in the figure legend needs to be mentioned. Sample number for each experiment not mentioned in any of the results. Three independent experiments were performed for all results. Based on the supplementary table of 37 samples, how were the samples divided for each independent experiment has not been mentioned.

Healthy UP

(n = 4)

CRSsNP-UP

(n = 10)

CRSwNP-UP (n = 10)

CRSwNP-NP (n = 13)

  • From the data presented, DNMT3A, a de novo methyl transferase seems to be the most effected by TGF-β1 and inhibitors. Why is that the case? It could be discussed in the discussion section.
  • In all results, the number of samples per experiment need to be mentioned. Was it pooled or individual samples that were tested is not clear?
  • Figure 2a. 1ng/ml TGF-β1 used DNMT3A most induced. The mRNA expression does not tie to what we see in protein expression, fig. 2b. DNMT1 and DNMT3B are higher as compared to DNMT3A for TGF-β1 stimulation in protein as compared to mRNA expression.
  • Figure 2c. The effect of TGF-β1 is more at 5 ng/ml which is closer to what is observed when 1ng/ml of TGF-β1 is used fig 2a. In figure 2c, 1.25 ng/ml does not have an effect like 1 ng/ml in fig 2a. Is it a biphasic effect based on concentration, especially for DNMT3A?
  • Likewise, a similar observation for protein in seen in Fig. 2d versus Fig 2b. 1.25 ng/ml seems to have a lesser effect than 1ng/ml. What is the concentration of protein loaded as the band intensity for Figure 2b looks different for fig. 2d even at the highest concentration of TGF-β1 5 ng/ml? A densitometric analysis of the bands normalized to the GADPH will be helpful here.
  • Figure 2. legend - #of replicates per sample (mRNA) and how many samples per experiment (mRNA and protein) were used to determine significance. What statistical test was used?
  • Figure 3a. Most impact seen for DNMT3A. Why was the choice made for 1ng/ml TGF-β1 here?  Also, why was 10 µM chosen to inhibit DNMT and why was this concentration used in the combinatorial effect with TGF-β1 has not been discussed. Fig. 3b pattern does not match with the mRNA expression as seen in Fig. 3a for the combination and individual effects for some of the DNMTs tested. Once again, sample numbers used per experiment not mentioned.
  • Figure 3d. the fibronectin protein pattern does not match with the mRNA expression pattern seen in Fig. 3c.

Minor concern

  • Sentence 55 needs rewording and space before 5
  • Sentence 63 it should be persistent
  • Sentence 94, although it is defined in the supplementary information, need to define CRSsNP and CRSwNP in the text where it first appears.
  • Figure 1. legend, although it is mentioned in the supplementary document, define or describe UP and NP in the text where it first occurs for ease of understanding.
  • Figure 4. Number of samples used per independent experiment. Statistical test used here?
  • Figure 5. Concentration of TGF-β1 used for stimulation after treatment with inhibitors.
  • Figure 6. TGF-β1concentration changed to 5 ng/ml explain. Protein loading concentrations for all blots.
  • Figure 7. TGF-β1 concentration changed to 1 ng/ml explain.
  • Sentence 281 explain further. Not clear how the data supports that.
  • Sentence 334, change "descrived" to described
  • In sentence 355, remove "each"
  • In sentence 384, need to mention protein concentration loaded
  • Sentences 391-393, mention the dilution factor for the respective antibodies used.
  • Sentence 427, remove “paste your materials and methods section here”.
  • Sentences 429-430, remove the sentence “For research articles with several authors, a short paragraph specifying their individual contributions must be provided. The following statements should be used”.
  • Sentences 434-436, remove “Please turn to the CRediT taxonomy for the term explanation. Authorship must be limited to those who have contributed substantially to the work reported”.

Author Response

Reviewer: 1
Comments to the Author 

1) In some experiments 1 ng/ml TGF-β1 was used and in others 5 ng/ml with no proper justification.

Response: This is my mistake. I used 5ng/ml of TGF-β1. In our previous study, we used TGF-β1 (5ng/ml) in A549 cells and primary nasal epithelial cell and in other paper, 5 ng/ml was used to induce EMT by TGF-β1 in A549 cells.

2) Statistical significance was obtained but the test used for each result in the figure legend needs to be mentioned. Sample number for each experiment not mentioned in any of the results. Three independent experiments were performed for all results. Based on the supplementary table of 37 samples, how were the samples divided for each independent experiment has not been mentioned.

Healthy UP

(n = 4)

CRSsNP-UP

(n = 10)

CRSwNP-UP

(n = 10)

CRSwNP-NP

(n = 13)

Response: In figure 1, we performed only one experiment using 37 samples mentioned in the supplementary table. Except for Figure 1, three independent experiments were performed in the remaining figures. So, we deleted the sentence “of three independent experiments.”.

3) From the data presented, DNMT3A, a de novo methyl transferase seems to be the most effected by TGF-β1 and inhibitors. Why is that the case? It could be discussed in the discussion section.

Response: As your comment, I added about DNMT3A

“We showed that DNMT3A seems to be the most effected by TGF-β1 and inhibitors. the modulation of DNMT3A, de novo DNA methyl transferase, is thought to have highly correlation with TGF-induced EMT process.”

4) In all results, the number of samples per experiment need to be mentioned. Was it pooled or individual samples that were tested is not clear?

Response: We performed three independent experiments using a single A549 cell line. We also performed three independent experiments using primary nasal epithelial cells isolated from three different CRS patients. So, we added this to each figure legend.

5) Figure 2a. 1ng/ml TGF-β1 used DNMT3A most induced. The mRNA expression does not tie to what we see in protein expression, fig. 2b. DNMT1 and DNMT3B are higher as compared to DNMT3A for TGF-β1 stimulation in protein as compared to mRNA expression.

Response: Thank you for your comment. I tried to confirm the protein expression of DNMT1, 3A, and 3B by various stimuli using western blot. We confirmed that DNMT3a was increased the most by TGF as shown in the mRNA data.

6) Figure 2c. The effect of TGF-β1 is more at 5 ng/ml which is closer to what is observed when 1ng/ml of TGF-β1 is used fig 2a. In figure 2c, 1.25 ng/ml does not have an effect like 1 ng/ml in fig 2a. Is it a biphasic effect based on concentration, especially for DNMT3A?

Response: As we mentioned in 1st question, it is my mistake. We used up to 5 ng/ml of TGF-β1.

7) Likewise, a similar observation for protein in seen in Fig. 2d versus Fig 2b. 1.25 ng/ml seems to have a lesser effect than 1ng/ml. What is the concentration of protein loaded as the band intensity for Figure 2b looks different for fig. 2d even at the highest concentration of TGF-β1 5 ng/ml? A densitometric analysis of the bands normalized to the GADPH will be helpful here.

Response: As we mentioned in 1st question, it is my mistake. we used up to 5 ng/ml of TGF-β1. And, we loaded 20 μg of protein.

8) Figure 2. legend - #of replicates per sample (mRNA) and how many samples per experiment (mRNA and protein) were used to determine significance. What statistical test was used?

Response: As I answer at 4th question, we performed three independent experiments using a single A549 cell line, and triplicates per sample. The statistical significance of differences between control and experimental data was analyzed using one-way analysis of variance (ANOVA) followed by Tukey's test.

9) Figure 3a. Most impact seen for DNMT3A. Why was the choice made for 1ng/ml TGF-β1 here?  Also, why was 10 µM chosen to inhibit DNMT and why was this concentration used in the combinatorial effect with TGF-β1 has not been discussed. Fig. 3b pattern does not match with the mRNA expression as seen in Fig. 3a for the combination and individual effects for some of the DNMTs tested. Once again, sample numbers used per experiment not mentioned.

Response: DNMT1, 3A, and 3B expression increased the most at 5 ng/ml of TGF-β1 (Figure 2c and 2d). 10uM of 5-Aza had no effect on cell viability, but cell viability decreased from 20 uM of 5-Aza (a). In addition to, 10 uM of 5-Aza inhibited both TGF-β1 induced DNMT expression and EMT inhibited the most (b and c).

10) Figure 3d. the fibronectin protein pattern does not match with the mRNA expression pattern seen in Fig. 3c.

Response: As you comment, I change the vimentin and fibronectin data.

Minor concern

1) Sentence 55 needs rewording and space before 5

Response: Thank you, I fixed it.

2) Sentence 63 it should be persistent

Response: Thank you, I change it.

3) Sentence 94, although it is defined in the supplementary information, need to define CRSsNP and CRSwNP in the text where it first appears.

Response: I added definition of CRSsNP and CRSwNP. “CRS without nasal polyp (CRSsNP) and CRS with nasal polyp (CRSwNP)”

4) Figure 1. legend, although it is mentioned in the supplementary document, define or describe UP and NP in the text where it first occurs for ease of understanding.

Response: I added definition of UP and NP in legend of figure 1.

“UP: Uncinate process, NP: Nasal polyp.”

5) Figure 4. Number of samples used per independent experiment. Statistical test used here?

Response: As answer of 8th question of major concern, we performed three independent experiments using a single A549 cell line, and triplicates per sample. The statistical significance of differences between control and experimental data was analyzed using one-way analysis of variance (ANOVA) followed by Tukey's test.

6) Figure 5. Concentration of TGF-β1 used for stimulation after treatment with inhibitors.

Response: In all data using inhibitors, we used 5ng/ml TGF-β1 after treatment with inhibitors.

7) Figure 6. TGF-β1 concentration changed to 5 ng/ml explain. Protein loading concentrations for all blots.

Response: As answer of 9th question of major concern, DNMT1, 3A, and 3B expression increased the most at 5 ng/ml of TGF-β1 (Figure 2c and 2d). 10uM of 5-Aza had no effect on cell viability, but cell viability decreased from 20uM of 5-Aza. In addition to, 10 uM of 5-Aza inhibited both TGF-β1 induced DNMT expression and EMT inhibited the most.

 We loaded 20μg of protein.

8) Figure 7. TGF-β1 concentration changed to 1 ng/ml explain.

Response: As we answer before, that is our mistake. We used TGF-β1 (1ng/ml) in Figure 7.

9) Sentence 281 explain further. Not clear how the data supports that.

Response: As your comment, I added the explain more.

“ERK1 and ERK2 are preferentially activated by mitogen stress, while p38 and JNK, members of the MAPK family, are activated by most stimuli such as inflammatory cytokines.”

10) Sentence 334, change "descrived" to described

Response: I fixed it.

11) In sentence 355, remove "each"

Response: I removed “each”.

12) In sentence 384, need to mention protein concentration loaded

Response: I added “(20 μg)”.

13) Sentences 391-393, mention the dilution factor for the respective antibodies used.

Response: I added dilution factor of antibodies “The primary antibodies included anti-DNMT1 (1:1000), anti-DNMT3A (1:1000), anti-DNMT3B (1: 1000, NOVUS Biologicals, Oakville, ON, Canada), anti-α-SMA (1:1000), (Abcam, Cambridge, MA, USA), anti-fibronectin (1:1000), anti-GAPDH (1:1000, Santa Cruz Biotechnology, Inc., Santa Cruz, CA, USA), anti-vimentin (1:1000), anti-Snail (1:1000) and anti-Slug (1:1000, Cell Signaling Technology, Danvers, MA, USA).”.

14) Sentence 427, remove “paste your materials and methods section here”.

Response: I deleted it.

15) Sentences 429-430, remove the sentence “For research articles with several authors, a short paragraph specifying their individual contributions must be provided. The following statements should be used”.

Response: I deleted it.

16) Sentences 434-436, remove “Please turn to the CRediT taxonomy for the term explanation. Authorship must be limited to those who have contributed substantially to the work reported”.

Response: I deleted it.

Reviewer 2 Report

Dear Authors,

please address the following issues:

  • Introduction. Please, explain better the limitations existing to treat CRS, as you made this statement in lines 43-45. Also, A549 are known as a lung carcinoma cell line. Therefore, please explain the function of this cell line in your experimental design. Also, the Lund-McKay score should be explained.
  • Line 55. There is a "5", possibly undesired. Please, rephrase the sentence in line 55 because it is not correct.
  • The names should be written in extenso the first time, with the abbreviation in brackets. What is the meaning of CRSsNP and CRSwNP written in line 94 ?
  • Figure 3. Please, remove immunoblottings of panel 3b because the quality is very low. The information is potentially interesting but the figure is not acceptable as it is.
  • Figure 4. Please, select better immunoblottings. The quality is still critical. I suggest pre-cast gels that yield a more homogeneous signal shape.
  • Figure 7a is very similar to figure 6c (the results of "control" and "TGFbeta1"). Figure 6e is very similar to figure 3a. I think that the upregulation of DNMTs by TGFbeta1 should be shown only once in figure 3. Thane the panel of figure 7a should be removed.
  • Discussion. The sentence "We hypothesized that DNMTs exert their effects through TRAF6 and TAK1 upstream of JNK and p38." it is not clear. Where are TRAF6 and TAK1 mentioned in the text ? Is the use of hypomethylating agents in the treatment of CRS already in clinical trials ?

Author Response

Reviewer: 2
Comments to the Author 

1) Introduction. Please, explain better the limitations existing to treat CRS, as you made this statement in lines 43-45. Also,

Response: I added the limitation existing to treat CRS. “At least 10% of patients with CRS develop recurrent and refractory disease that does not respond to medical and surgical treatment regimens.”

2) A549 are known as a lung carcinoma cell line. Therefore, please explain the function of this cell line in your experimental design.

Response: As a limitation of this study, it has been added to the discussion section.

“It is a limitation that we confirmed the above mechanism using A459, and confirmed it using primary nasal epithelial cells. Because of the difficulty in obtaining the relevant target organ tissues and the difficulty in culturing them, most upper respiratory epithelial studies have relied heavily on commercially produced cells or transformed cell lines, including A549.”

3) Also, the Lund-McKay score should be explained.

Response: I added the explain of the Lund-McKay score in result 2.1.

“The Lund-Mackay score is a widely used method for radiological staging of chronic rhinosinusitis by CT scan of the sinus and periosteum complex. The Lund-MacKay score expresses the opacity of the bilateral sinuses on a scale of 0 to 24.”

4) Line 55. There is a "5", possibly undesired. Please, rephrase the sentence in line 55 because it is not correct.

Response: I deleted it. Thank you.

5) The names should be written in extenso the first time, with the abbreviation in brackets. What is the meaning of CRSsNP and CRSwNP written in line 94 ?

 Response: I added definition of CRSsNP and CRSwNP. “CRS without nasal polyp (CRSsNP) and CRS with nasal polyp (CRSwNP)”

6) Figure 3. Please, remove immunoblottings of panel 3b because the quality is very low. The information is potentially interesting but the figure is not acceptable as it is.

Response: As your comment, I re-confirmed panel 3b data using western blot.

7) Figure 4. Please, select better immunoblottings. The quality is still critical. I suggest pre-cast gels that yield a more homogeneous signal shape.

Response: As your comment, I re-confirmed Figure 4. data using western blot.

8) Figure 7a is very similar to figure 6c (the results of "control" and "TGFbeta1"). Figure 6e is very similar to figure 3a. I think that the upregulation of DNMTs by TGFbeta1 should be shown only once in figure 3. Thane the panel of figure 7a should be removed.

Response: The results of Figure 7a and Figure 6a may look similar, but Figure 7a is the result using primary nasal epithelial cells and Figure 6a is the result using a549 cells. Therefore, the meaning is different.

9) Discussion. The sentence "We hypothesized that DNMTs exert their effects through TRAF6 and TAK1 upstream of JNK and p38." it is not clear. Where are TRAF6 and TAK1 mentioned in the text ?

Response: We agree with your comment. We show only JNK and p38 signaling pathways for DNMT regulation. So, we deleted “TRAF6 and TAK upstream of”

10) Is the use of hypomethylating agents in the treatment of CRS already in clinical trials ?

Response: We know that hypomethylating agents have not yet been used in clinical trials for the treatment of CRS.

Round 2

Reviewer 2 Report

Dear Authors,

I saw the improvements to the manuscript.